# Solidification Behavior of Dy-Tb-Fe Alloys through Experimental Study and Thermodynamic Calculation

**DOI:** 10.3390/ma16134697

**Published:** 2023-06-29

**Authors:** Cong Tan, Qi Wei, Weifeng Cheng, Xingyu Liu, Yuchen Bai, Jiang Wang

**Affiliations:** 1Guangxi Key Laboratory of Information Materials, School of Materials Science and Engineering, Guilin University of Electronic Technology, Guilin 541004, China; tancong0526@163.com (C.T.); 17773055566@163.com (Q.W.); m17828906669@163.com (W.C.); liuxingyu0026@163.com (X.L.); 13124132992@163.com (Y.B.); 2Engineering Research Center of Electronic Information Materials and Devices, Ministry of Education, Guilin University of Electronic Technology, Guilin 541004, China

**Keywords:** Dy-Tb-Fe, solidification microstructure, phase transition, thermodynamic calculation

## Abstract

In this work, the solidification microstructure and phase transitions of Dy-Tb-Fe alloy samples were studied by using scanning electron microscopy with energy dispersive spectroscopy (SEM-EDS), X-ray diffraction (XRD) and differential thermal analysis (DTA). No stable ternary compound was detected in the present experiments. The phase transformation temperatures of eight Dy-Tb-Fe alloy samples were measured. Based on the experimental results determined in this work and reported in the literature, the phase equilibria of the Dy-Tb-Fe system was calculated using the CALPHAD method. The calculated vertical sections are consistent with the experimental results determined in this work and reported in the literature. Furthermore, in combination with the experimental solidification microstructure, the solidification behavior of Dy-Tb-Fe alloy samples was analyzed through the thermodynamic calculation with the Gulliver–Scheil non-equilibrium model. The simulated results agree well with the experimental results. This indicates that the reasonable thermodynamic parameters of the Dy-Tb-Fe system were finally obtained.

## 1. Introduction

Nd-Fe-B permanent magnets with excellent magnetic properties have been widely used in various industrial fields such as wind turbines, electric vehicles, and aerospace [1,2,3,4,5]. With the increase in the operation temperature, the magnetic properties (e.g., coercivity and remanence) of Nd-Fe-B permanent magnets decrease [6,7,8,9]. In order to ensure the sufficient coercivity of Nd-Fe-B permanent magnets at the operation temperature, the addition of heavy rare earth elements (Dy, Tb, etc.) to Nd-Fe-B permanent magnets to partially substitute Nd is an effective method because the magnetocrystalline anisotropy fields of Dy_2_Fe_14_B and Tb_2_Fe_14_B are much higher than that of Nd_2_Fe_14_B, which would result in the great improvement of the coercivity for the permanent magnets [10,11,12]. To reduce the content of heavy rare earth metals Dy and Tb in Nd-Fe-B permanent magnets, the grain boundary diffusion process (GBDP) was developed recently [13]. During the GBDP, first, Dy and Tb can diffuse along the grain boundary phase into the interior of the magnet; then, the partial Nd on the surface of Nd_2_Fe_14_B grains is replaced by Dy and Tb to form a core–shell microstructure containing Nd_2_Fe_14_B and (Dy, Tb)_2_ Fe_14_B. The substitution of Nd by Dy and Tb dilutes the ferromagnetism of the grain boundary phase and further weakens the coupling between neighboring grains, which enhances the coercivity of the magnets after the grain boundary diffusion process [14,15]. In order to gain a deeper understanding of the grain boundary diffusion process, the thermodynamics and kinetics of Nd-Fe-B permanent magnets containing Tb and Dy elements are fundamental to enhance the overall performance of Nd-Fe-B magnets [16]. In particular, the solidification process of Nd-Fe-B-based alloys is significantly important to control microstructure and improve magnetic properties of Nd-Fe-B permanent magnets [17]. Solidification is a process in which the liquid phase decreases gradually and the solid phase is formed by peritectic, eutectic and other reactions during the cooling process until the liquid phase disappears completely [18]. Using thermodynamic calculations, the Scheil–Gulliver model can be employed to simulate the non-equilibrium solidification process of as-cast alloys [19]. In our previous work, the RE-Fe (RE = Pr, Nd, Sm, Gd, Dy, Tb, Ho, Tm, Lu, Y) [20,21,22,23,24], RE-B (RE = La, Ce, Pr, Nd, Sm, Gd, Dy, Tb, Ho, Tm, Lu, Y) [25,26,27], and RE_1_-RE_2_ binary systems [28] were calculated, and then thermodynamic calculations of the RE_1_-RE_2_-Fe (e.g., La-Ce-Fe and Ce-Nd-Fe [29], La-Pr-Fe and Ce-Pr-Fe [30]) ternary systems and the Nd-Fe-B ternary system [31] were performed. Furthermore, the development of a thermodynamic database of Nd-Dy-Tb-Fe-B magnets is in progress in our group.

In this work, the solidification microstructure and phase transitions of Dy-Tb-Fe as-cast alloy samples were investigated experimentally, and then phase equilibria of this ternary system was calculated using the CALPHAD method. Finally, the solidification behaviors of Dy-Tb-Fe as-cast alloy samples were investigated using the Scheil–Gulliver non-equilibrium model. It could demonstrate a solid foundation for the microstructure design of high-performance Nd-Fe-B permanent magnet materials with Dy and Tb elements.

## 2. Literature Information

### 2.1. Binary Systems

The thermodynamic database of Dy-Fe and Tb-Fe systems in the high-temperature range (above 800 K) was established by Landin et al. [32], but the contribution of magnetism to the Gibbs energy and experimental heat capacity of all intermetallic compounds were not taken into account. Rong et al. [22] optimized the Dy-Fe and Tb-Fe systems using the CALPHAD method considering the magnetic contribution and experimental heat capacity of the intermetallic compounds. The calculation results including phase relationship and thermodynamic properties are in good agreement with the experimental results. Recently, Ye et al. [24] re-conducted thermodynamic calculations for Dy-Fe and Tb-Fe systems to achieve compatibility with the RE-Fe thermodynamic database. The results of the Dy-Fe and Tb-Fe systems optimized by Ye et al. [24] were used in the present calculation of the Dy-Tb-Fe system.

The Dy-Tb phase diagram was not calculated in the reported literature up to now. Gschneidner et al. [33,34] reported that Dy and Tb are completely miscible and measured lattice parameters of the Dy-Tb alloys in the systematic review of lanthanide binary systems. According to the reported data [33,34], the Dy-Tb phase diagram was drawn by Moffatt [35] considering the continuous solid solution phases formed from β-Dy, β-Tb, α-Dy and α-Tb phases due to their same crystal structures [36]. Therefore, all the phases including liquid phase, bcc (β-Dy, β-Tb) and hcp (α-Dy, α-Tb) in the Dy-Tb system were described by using the ideal solution model. Figure 1 shows the calculated Dy-Tb phase diagram in this work.

### 2.2. Ternary System

The Dy_0.73_Tb_0.27_-Fe vertical section of the Dy-Tb-Fe system was measured by Westwood et al. [37] with DTA, X-ray diffraction and metallography. Landin et al. [32] directly extrapolated the Dy-Tb-Fe system based on the experimental results of Westwood et al. and Abell et al. [38], including the Dy_0.73_Tb_0.27_-Fe vertical section and liquidus projection. Although the values of the calculated Dy_0.73_Tb_0.27_-Fe vertical section are consistent with the experimental data [37], Ye et al. [24] have recently reassessed the calculations of the Dy-Fe and Tb-Fe systems. Therefore, thermodynamic calculations for the Dy-Tb-Fe system are still necessary.

## 3. Experimental Procedure

Eight Dy-Tb-Fe alloy samples with compositions of $x_{Dy}:x_{Tb}=1:1$ were prepared from bulk Dy, Tb and Fe (99.99% purity, China New Metal Material Technology Company, Ltd., Beijing, China). Each alloy sample (about 4 g) was melted 3–5 times in a vacuum arc melting furnace filled with argon gas to ensure uniform composition. The alloy samples were cooled in a copper crucible with cooling water, and thus the solidification process of the alloy samples was non-equilibrium due to the fast cooling rate.

The as-cast alloy samples were prepared by standard metallographic procedure. The morphology and phase composition of the alloy samples were measured by scanning electron microscope with energy dispersive spectroscopy (SEM-EDS, FEI 450G, FEI Company, Hillsboro, OR, USA). The compositions of each phase in the alloy samples were measured four times by EDS, and the standard deviations of the measured composition data were determined. After the alloy sample was ground into powder in anhydrous ethanol, the phase structures of the formed phases in the alloy samples were analyzed by X-ray powder diffraction (XRD, PLXcel 3D, Cu K_α_ radiation). The phase transition temperatures of the alloy samples were measured by differential thermal analysis (DTA, TA Instruments SDT/Q-600) using high-purity Al_2_O_3_ crucibles in a flowing argon atmosphere. Considering that rare earth metals are prone to oxidation, a heating/cooling rate of 20 K/min was used in DTA measurement.

## 4. Thermodynamic Calculation

### 4.1. Solution Phases

The solution phase *φ* including liquid, fcc, bcc and hcp is described by using the substitutional solution model. The molar Gibbs energy of the solution phase *φ* can be expressed as follows:(1)Gmφ=∑i=Dy,Tb,FexiGiφ0+RT∑i=Dy,Tb,Fexilnxi + Gmφmag+Gmφex,
(2)Gmφ = exxDyxTb∑j=0LDy,TbφxDy−xTbjj+xDyxFe∑j=0LDy,FeφxDy−xFejj,+xTbxFe∑j=0LTb,FeφxTb−xFejj+xDyxTbxFeLDy,Tb,Feφ,
(3)Gmφmag=RTlnβ0+1gτ,
(4)LDy,Tb,Feφ=xDyLDy,Tb,Feφ + 0xTbLDy,Tb,Feφ + 1xFeLDy,Tb,Feφ2,
where ${x_{i}}^{}$ is the mole fraction of element *i* (*i* = Dy, Tb, Fe) and Giφ0 means the molar Gibbs energy of phase *φ* for element *i* (*i* = Dy, Tb, Fe); these values refer to the SGTE database [39]. *R* is the gas constant and *T* is the absolute temperature (Kelvin). Gmφmag is the magnetic contribution to Gibbs energy of the magnetic phase. In Equation (3), $\tau =T/T_{c}^{\varphi }$, and $T_{c}^{\varphi }$ is the Curie temperature of the phase *φ*. $\beta_{0}$ is the Bohr magnetrons. LDy,Feφj and LTb,Feφj are the interaction parameters and are taken from the Dy-Fe and Tb-Fe systems evaluated by Ye et al. [24], respectively. LDy,Tb,Feφ0, LDy,Tb,Feφ1 and LDy,Tb,Feφ2 are the ternary interaction parameters to be evaluated.

### 4.2. Intermetallic Compounds

In the Dy-Tb-Fe system, a continuous solid solution is formed because of the same crystal structure of DyFe_2_ and TbFe_2_ [40,41,42]. Similarly, DyFe_3_ and TbFe_3_, Dy_6_Fe_23_ and Tb_6_Fe_23_, Dy_2_Fe_17_ and Tb_2_Fe_17_ also form a continuous solid solution in the Dy-Tb-Fe system. Therefore, these intermetallic compounds are modeled by (Dy, Tb)_0.3333_Fe_0.66667_, (Dy, Tb)_0.25_Fe_0.75_, (Dy, Tb)_0.2069_Fe_0.7931_ and (Dy, Tb)_0.1053_Fe_0.8947_, which are named REFe_2_, REFe_3_, RE_6_Fe_23_ and RE_2_Fe_17_ (RE = Dy, Tb). Their molar Gibbs energies can be expressed as
(5)GmREFe2=YDy GDy:FeREFe2+YTb GTb:FeREFe2 + 0.3333RTYDylnYDy + YTblnYTb + YDyYTbLDy,Tb:FeREFe2xDy−xTbjj + GmREFe2mag,
(6)LDy,Tb:FeREFe2j=A1+B1T,
(7)GmREFe3=YDyGDy:FeREFe3 + YTbGTb:FeREFe3 + 0.25RTYDylnYDy + YTblnYTb + YDyYTbLDy,Tb:FeREFe3xDy−xTbjj + GmREFe3mag,
(8)LDy,Tb:FeREFe3j=A2+B2T,
(9)GmRE6Fe23=YDyGDy:FeRE6Fe23 + YTbGTb:FeRE6Fe23 + 0.2069RTYDylnYDy+YTblnYTb + YDyYTbLDy,Tb:FeRE6Fe23xDy−xTbjj + GmRE6Fe23mag,
(10)LDy,Tb:FeRE6Fe23j=A3+B3T,
(11)GmRE2Fe17=YDyGDy:FeRE2Fe17 + YTbGTb:FeRE2Fe17 + 0.1053RTYDylnYDy + YTblnYTb + YDyYTbLDy,Tb:FeRE2Fe17xDy−xTbjj + GmRE2Fe17mag,
(12)LDy,Tb:FeRE2Fe17j=A4+B4T,
in which LDy:FeREFe2j, LTb:FeREFe2j, LDy:FeREFe3j, LTb:FeREFe3j, LDy:FeRE6Fe23j, LTb:FeRE6Fe23, jLDy:FeRE2Fe17j and LTb:FeRE2Fe17j are interaction parameters taken from the Dy-Fe and Tb-Fe systems assessed by Ye et al. [24]. LDy,Tb:FeREFe2j, LDy,Tb:FeREFe3j, LDy,Tb:FeRE6Fe23j and LDy,Tb:FeRE2Fe17j are the interaction parameters to be optimized.

## 5. Results and Discussion

The microstructure and phase transitions of eight Dy-Tb-Fe as-cast alloy samples were determined in this work. The phase compositions and phase transition temperatures of the alloy samples measured by EDS, XRD and DTA are shown in Table 1.

### 5.1. Solidification Microstructure

Figure 2 contains the BSE micrograph and XRD patterns of Dy_42.5_Tb_42.5_Fe_15_ alloy sample. In Figure 2a, the microstructure of this sample shows the formation of two phases, and the composition of the gray phase was measured by EDS to be 17.88 at.% Dy, 13.91 at.% Tb and 68.21 at.% Fe, while that of the light gray phase was determined to be 46.77 at.% Dy, 47.74 at.% Tb and 5.49 at.% Fe. According to the results determined by EDS in Table 1, the gray phase and the light gray phase were identified to be (Dy, Tb) Fe_2_ and hcp (Dy, Tb), respectively, which was same as those of the XRD patterns in Figure 2b. Moreover, there is a large amount of the hcp (Dy, Tb) phase in Figure 2b, and the background of the diffraction pattern is too high and the spectral peak is not smooth, which is the result of the internal stress or preferred orientation generated during the preparation of metal powders. In addition, the microstructure characteristics of the Dy_42.5_Tb_42.5_Fe_15_ alloy sample indicate that the light gray hcp (Dy, Tb) phase was formed first from the liquid phase during the solidification process.

Figure 3 shows the BSE and XRD images of Dy_25_Tb_25_Fe_50_ and Dy_20_Tb_20_Fe_60_ alloy samples. In Figure 3a,c, Dy_25_Tb_25_Fe_50_ and Dy_20_Tb_20_Fe_60_ alloy samples are composed of two phases, while their XRD patterns demonstrate the formation of the (Dy, Tb) Fe_2_ and hcp (Dy, Tb) phase in Figure 3b,d. According to EDS results, there are two phases present in these two samples, with (Dy, Tb) Fe_2_ in dark gray and hcp (Dy, Tb) in light gray. It means that the SEM-EDS results of these two samples are consistent with their XRD results. Meanwhile, the formation of the primary phase (Dy, Tb) Fe_2_ and similar eutectic microstructure including (Dy, Tb) Fe_2_ and hcp (Dy, Tb) was observed from the BSE micrographs of these two as-cast alloy samples.

As shown in Figure 4a,c, Dy_15_Tb_15_Fe_70_ and Dy_13.5_Tb_13.5_Fe_73_ alloy samples present the microstructure of two phases. Based on the composition measurements in Table 1, these two phases are identified to be (Dy, Tb) Fe_2_ and (Dy, Tb) Fe_3_, which are in good agreement with the XRD patterns in Figure 4b,d. Similarly, the microstructure characteristics of Dy_15_Tb_15_Fe_70_ and Dy_13.5_Tb_13.5_Fe_73_ alloy samples show that the (Dy, Tb) Fe_3_ phase is the primary phase during this solidification process.

Figure 5 displays the BSE micrographs and XRD patterns of Dy_12_Tb_12_Fe_76_ and Dy_10.5_Tb_10.5_Fe_79_ alloy samples. In Figure 5a,c, three different phases were formed in these two samples. As given in Table 1, the experimental results obtained by EDS indicate that the light gray phase, the gray phase, and the dark black phase are (Dy, Tb) Fe_3_, (Dy, Tb)_6_ Fe_23_ and (Dy, Tb)_2_ Fe_17_, respectively, which are same as those of the XRD patterns in Figure 5b,d. The microstructures of Dy_12_Tb_12_Fe_76_ and Dy_10.5_Tb_10.5_Fe_79_ alloy samples suggest that the (Dy, Tb)_6_Fe_23_ phase as the primary phase was formed.

Figure 6 presents the BSE and XRD images of the Dy_9_Tb_9_Fe_82_ alloy sample. Based on the microstructure and XRD patterns in Figure 6a,b with the phase compositions measured by EDS in Table 1, the formation of the (Dy, Tb)_6_ Fe_23_ phase and the (Dy, Tb)_2_ Fe_17_ phase was found. Moreover, (Dy, Tb)_2_ Fe_17_ was formed as the primary phase in the Dy_9_Tb_9_Fe_82_ alloy sample.

Based on the experimental results of eight Dy-Tb-Fe alloy samples with the compositions of $x_{Dy}:x_{Tb}=1:1$ determined by SEM-EDS and XRD, it was noted that the stable ternary intermetallic compound was not determined in this work. In addition, the EDS results measured in Table 1 illustrate that the solubilities of Dy in TbFe_2_, TbFe_3_, Tb_6_Fe_23_ and Tb_2_Fe_17_ as well as those of Tb in DyFe_2_, DyFe_3_, Dy_6_Fe_23_ and Dy_2_Fe_17_ are different. This indicates that (Dy, Tb) Fe_2_, (Dy, Tb) Fe_3_, (Dy, Tb)_6_Fe_23_ and (Dy, Tb)_2_ Fe_17_ all form continuous solid solution phase in the Dy-Tb-Fe system, which was also reported by Westwood et al. [37].

### 5.2. Phase Transition

Figure 7 shows the thermal analysis curve of Dy-Tb-Fe alloy samples in this work. Based on the thermal analysis results, the transition temperatures of Dy-Tb-Fe alloy samples were analyzed, and the results are listed in Table 1. In Figure 7a, the thermal curve of the Dy_42.5_Tb_42.5_Fe_15_ alloy sample displays three peaks at 1030 K, 1057 K and 1435 K. Combined with Figure 2a, the peaks at 1030 K and 1057 K are generated due to the formation of the hcp (Dy, Tb) phase and the (Dy, Tb) Fe_2_ phase, while the third peak at 1435 K is corresponding to the formation of the hcp (Dy, Tb) phase as the primary phase. The thermal curve of the Dy_25_Tb_25_Fe_50_ alloy sample in Figure 7b presents three signal peaks at 1121 K, 1131 K and 1459 K, respectively, which correspond to the formation of the (Dy, Tb) Fe_2_ and the hcp (Dy, Tb) phase. Nevertheless, the thermal curve of the Dy_20_Tb_20_Fe_60_ alloy sample in Figure 7c shows only one endothermic peak at 1490 K, corresponding to the formation of (Dy, Tb) Fe_2_, although the solidification microstructures of both Dy_25_Tb_25_Fe_50_ and Dy_20_Tb_20_Fe_60_ alloy samples contain the (Dy, Tb) Fe_2_ and hcp (Dy, Tb) phases. In Figure 7d,e, the thermal analysis curves of Dy_15_Tb_15_Fe_70_ and Dy_13.5_Tb_13.5_Fe_73_ alloy samples indicate two peaks at 1494/1475 K and 1501/1505 K, corresponding to the formation of (Dy, Tb) Fe_2_ and (Dy, Tb) Fe_3_. Similarly, the thermal analysis curve of the Dy_12_Tb_12_Fe_76_ alloy sample in Figure 7f indicates three signal peaks at 1482 K, 1495 K and 1555 K, corresponding to the formation of (Dy, Tb) Fe_3_, (Dy, Tb)_6_ Fe_23_ and (Dy, Tb)_2_ Fe_17_. However, two peaks at 1497 K and 1562 K were observed in the DTA results of the Dy_10.5_Tb_10.5_Fe_79_ alloy sample in Figure 7g, although the solidification microstructures of both Dy_12_Tb_12_Fe_76_ and Dy_10.5_Tb_10.5_Fe_79_ alloy samples consist of (Dy, Tb) Fe_3_, (Dy, Tb)_6_ Fe_23_ and (Dy, Tb)_2_ Fe_17_. As shown in Figure 7h, two endothermic peaks at 1523 K and 1560 K were observed in the Dy_9_Tb_9_Fe_82_ alloy sample, which correspond to the formation of (Dy, Tb)_6_ Fe_23_ and (Dy, Tb)_2_ Fe_17_. In addition, the oxidation peaks due to the easy oxidation of Dy-Tb-Fe alloy samples at high temperatures were also observed in Figure 7f,h.

### 5.3. Thermodynamic Calculation

Based on the experimental data determined in this work and reported by Westwood et al. [37] as well as previous evaluations of Dy-Fe, Tb-Fe, and Tb-Dy systems, the Dy-Tb-Fe system was calculated. The thermodynamic parameters of the Dy-Tb-Fe system obtained are listed in Table 2.

Figure 8 shows the calculated liquidus projection of the Dy-Tb-Fe system. It was found that no invariant reactions are existent in this ternary system. The calculated liquidus projection agrees with the experimental results of the primary phase determined in the experiment. Figure 9 shows the vertical sections of Tb_0.50_Dy_0.50_-Fe and Tb_0.27_Dy_0.73_-Fe calculated based on the experimental data determined in this work and reported by Westwood et al. [37]. The calculation results differ slightly from the experimental results and are still accepted within the experimental errors considering the oxidation of Dy-Tb-Fe alloy samples in the thermal analysis measurements at high temperature.

In order to gain a deeper understanding of the phase transformation of Dy-Tb-Fe alloy samples during the non-equilibrium solidification process, the thermodynamic parameters of the Dy-Tb-Fe system obtained in this work were employed to simulate the solidification process of the as-cast alloy samples using the Gulliver–Scheil model. Figure 10 displays the solidification process of four alloy samples (Dy_42.5_Tb_42.5_Fe_15_, Dy_15_Tb_15_Fe_70_, Dy_10.5_Tb_10.5_Fe_79_, and Dy_9_Tb_9_Fe_82_) simulated using the Gulliver–Scheil module. The simulated solidification path of the Dy_42.5_Tb_42.5_Fe_15_ alloy sample in Figure 10a is as follows: L → hcp (Dy, Tb) + L → hcp (Dy, Tb) +L + (Dy, Tb) Fe_2_. The calculated results show that the solidification structure of the Dy_42.5_Tb_42.5_Fe_15_ alloy sample consists of a hcp (Dy, Tb) phase and (Dy, Tb) Fe_2_, which is consistent with the microstructure observation results shown in Figure 2a. Figure 10b shows the simulated solidification path of the Dy_15_Tb_15_Fe_70_ alloy sample: L → (Dy, Tb) Fe_3_ + L → (Dy, Tb) Fe_3_ + L + (Dy, Tb) Fe_2_. It indicates that the microstructure of the Dy_15_Tb_15_Fe_70_ alloy sample consists of (Dy, Tb) Fe_3_ and (Dy, Tb) Fe_2_. The simulation results are the same as the SEM results shown in Figure 4a. Similarly, Figure 10c shows the simulated solidification path of the Dy_10.5_Tb_10.5_Fe_79_ alloy sample: L → L + (Dy, Tb)_6_Fe_23_ → L + (Dy, Tb)_6_ Fe_23_ + (Dy, Tb) Fe_3_, while the simulated solidification path of the Dy_9_Tb_9_Fe_82_ alloy sample in Figure 10d is as follows: L → L + (Dy, Tb)_2_Fe_17_ → L + (Dy, Tb)_2_Fe_17_ + (Dy, Tb)_6_Fe_23_, which is consistent with the microstructure observation results shown in Figure 5c and Figure 6a. This indicates that reliable thermodynamic data of the Dy-Tb-Fe system was obtained in this work, which can be employed to reproduce well the solidification processes of Dy-Tb-Fe alloy samples using the Gulliver–Scheil module.

## 6. Conclusions

This work investigates the solidification behavior of the Dy-Tb-Fe system using experimental measurements and thermodynamic calculations. The conclusions drawn are as follows:The phase transition temperatures and phase compositions of eight Dy-Tb-Fe alloy samples were determined. Based on the experimental results determined in this work and reported in the literature, the thermodynamic calculation of the Dy-Tb-Fe system was performed using the CALPHAD method. The calculated vertical section and liquidus projection are consistent with the experimental results.The solidification behaviors of several Dy-Tb-Fe alloy samples were simulated by using the Gulliver–Scheil non-equilibrium model with the obtained thermodynamic parameters. The simulation results were compared with the solidification structure of the experimental samples, and they were in good agreement with the experimental results. This means that the thermodynamic parameters of the Dy-Tb-Fe system optimized in this work are reliable and will provide reference for the microstructure design of high-performance Nd-Dy-Tb-Fe-B magnets.

## Figures and Tables

**Figure 1 materials-16-04697-f001:**
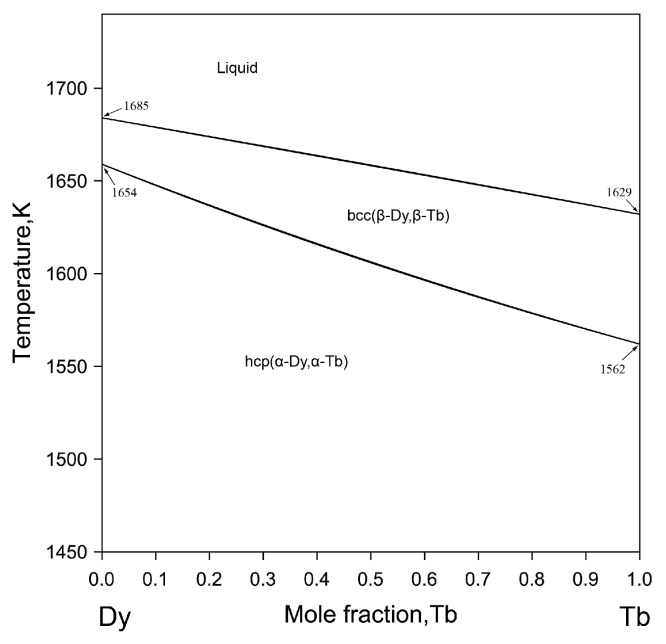
The calculated Dy-Tb binary phase diagram in this work.

**Figure 2 materials-16-04697-f002:**
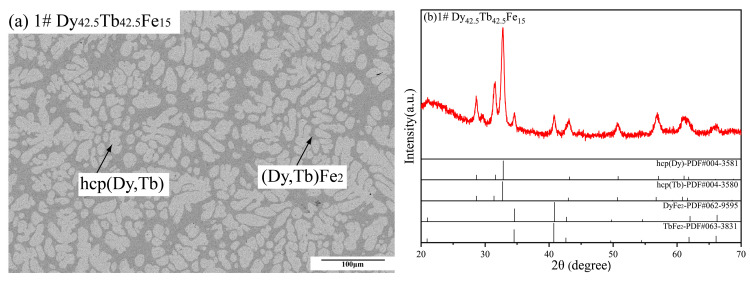
(**a**) Back-scattered electron (BSE) micrograph and (**b**) XRD patterns of Dy_42.5_Tb_42.5_Fe_15_ as-cast alloy sample.

**Figure 3 materials-16-04697-f003:**
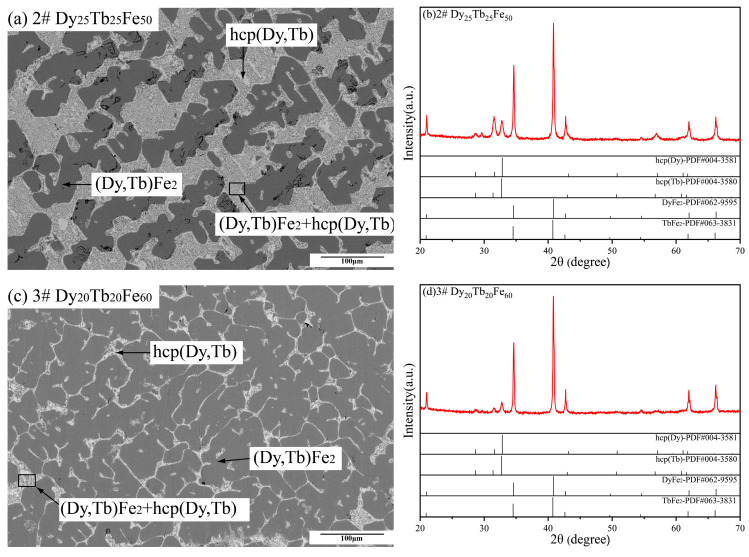
Back-scattered electron (BSE) micrographs and XRD patterns of Dy_25_Tb_25_Fe_50_ and Dy_20_Tb_20_Fe_60_ as-cast alloy samples. (**a**,**b**) Dy_25_Tb_25_Fe_50_, (**c**,**d**) Dy_20_Tb_20_Fe_60_.

**Figure 4 materials-16-04697-f004:**
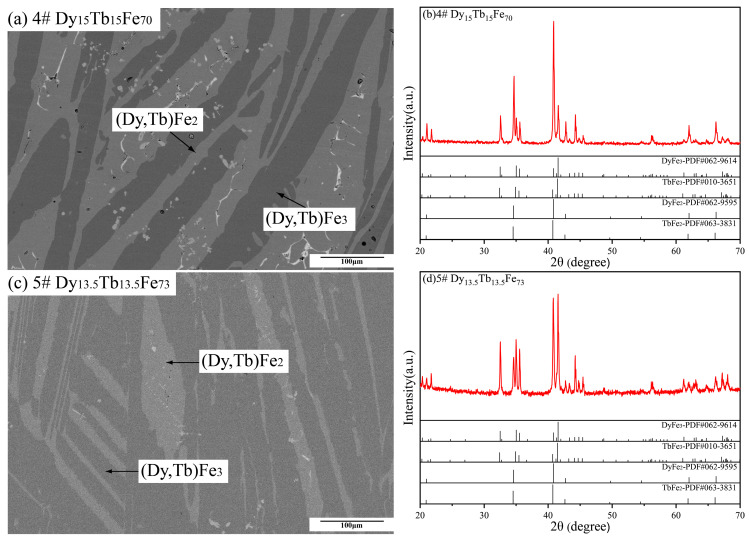
Back-scattered electron (BSE) micrographs and XRD patterns of Dy_15_Tb_15_Fe_70_ and Dy_13.5_Tb_13.5_Fe_73_ as-cast alloy samples. (**a**,**b**) Dy_15_Tb_15_Fe_70_, (**c**,**d**) Dy_13.5_Tb_13.5_Fe_73_.

**Figure 5 materials-16-04697-f005:**
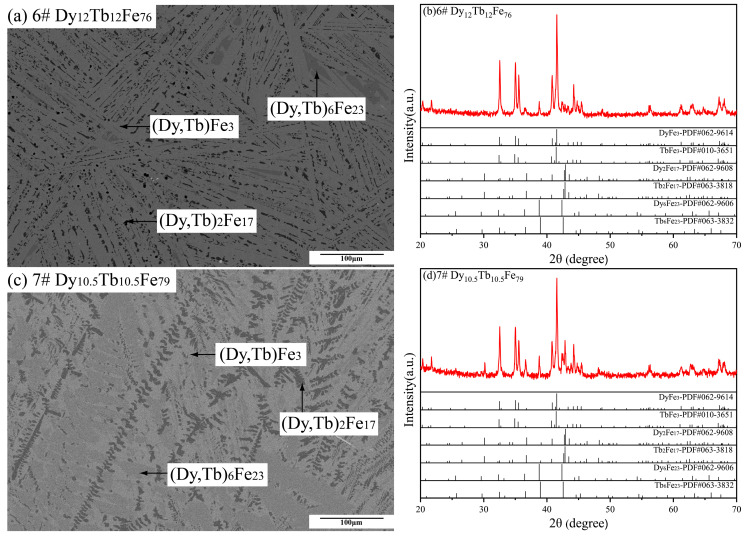
Back-scattered electron (BSE) micrographs and XRD patterns of Dy_12_Tb_12_Fe_76_ and Dy_10.5_Tb_10.5_Fe_79_ as-cast alloy samples. (**a**,**b**) Dy_12_Tb_12_Fe_76_, (**c**,**d**) Dy_10.5_Tb_10.5_Fe_79_.

**Figure 6 materials-16-04697-f006:**
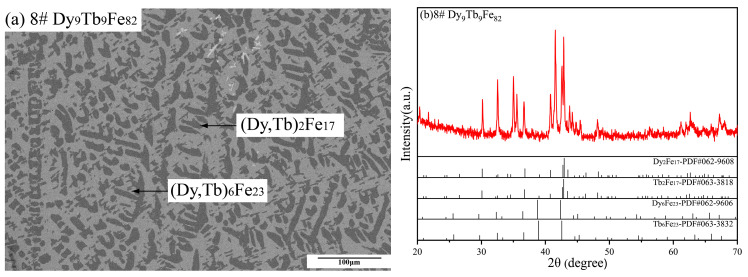
(**a**) Back-scattered electron (BSE) micrographs and (**b**) XRD patterns of Dy_9_Tb_9_Fe_82_ as-cast alloy sample.

**Figure 7 materials-16-04697-f007:**
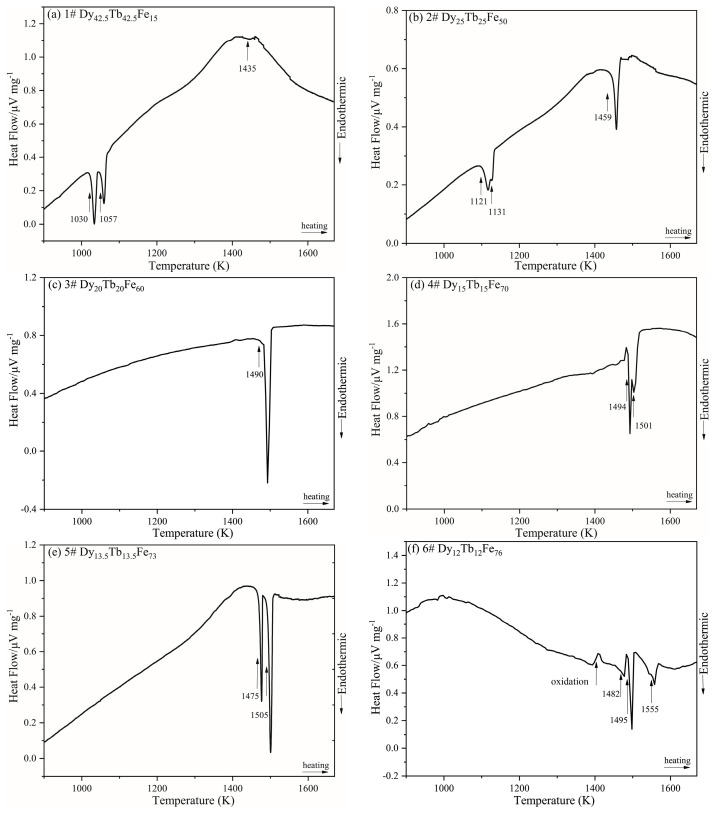
Thermal analysis curves of Dy-Tb-Fe alloy samples measured by DTA at heating rate of 20 K/min. (**a**) Dy_42.5_Tb_42.5_Fe_15_, (**b**) Dy_25_Tb_25_Fe_50_, (**c**) Dy_20_Tb_20_Fe_60_, (**d**) Dy_15_Tb_15_Fe_70_, (**e**) Dy_13.5_Tb_13.5_Fe_73_, (**f**) Dy_12_Tb_12_Fe_76_, (**g**) Dy_10.5_Tb_10.5_Fe_79_, (**h**) Dy_9_Tb_9_Fe_82_.

**Figure 8 materials-16-04697-f008:**
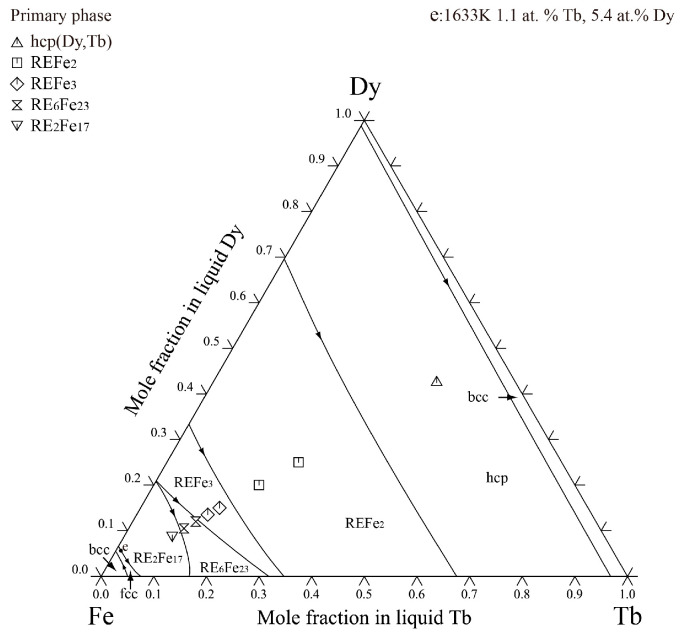
The calculated liquidus projection of the Dy-Tb-Fe system with the experimental results determined in this work.

**Figure 9 materials-16-04697-f009:**
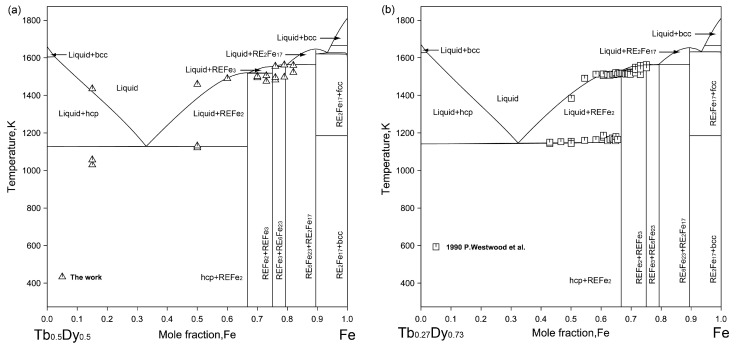
The calculated vertical sections under the different composition conditions with the experimental data measured in this work and reported by Westwood et al. [37]. (**a**) $x_{Dy}:x_{Tb}=1:1$, (**b**) $x_{Dy}:x_{Tb}=0.73:0.27$.

**Figure 10 materials-16-04697-f010:**
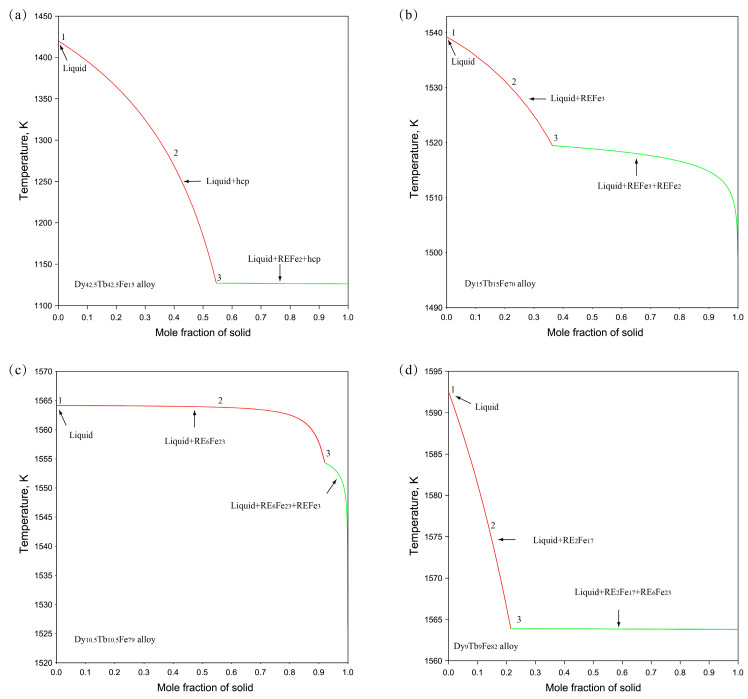
Calculated Scheil–Gulliver solidification curves of the representative Dy-Tb-Fe as-cast alloy samples. (**a**) Dy_42.5_Tb_42.5_Fe_15_, (**b**) Dy_15_Tb_15_Fe_70_, (**c**) Dy_10.5_Tb_10.5_Fe_79_, (**d**) Dy_9_Tb_9_Fe_82_.

**Table 1 materials-16-04697-t001:** Phase compositions and identified phases of Dy-Tb-Fe alloy samples.

Nominal Composition (at.%)	SEM-EDS Results (at.%)			Identified Phases by EDS and XRD	Phase Transitions (K)	Solidification Process
	Dy	Tb	Fe			
1#-Dy_42.5_Tb_42.5_Fe_15_	46.77 ± 0.2	47.74 ± 0.6	5.49 ± 0.4	hcp (Dy, Tb)	1030, 1057, 1435	L → hcp (Dy, Tb) L → hcp (Dy, Tb) + (Dy, Tb) Fe_2_
	17.88 ± 0.6	13.91 ± 0.2	68.21 ± 0.4	(Dy, Tb) Fe_2_		
2#-Dy_25_Tb_25_Fe_50_	17.51 ± 0.7	15.46 ± 0.3	67.03 ± 0.6	(Dy, Tb) Fe_2_	1121, 1131, 1459	L → (Dy, Tb) Fe_2_ L → hcp (Dy, Tb) + (Dy, Tb) Fe_2_
	40.64 ± 0.2	52.09 ± 0.1	7.27 ± 0.2	hcp (Dy, Tb)		
3#-Dy_20_Tb_20_Fe_60_	17.33 ± 0.7	15.96 ± 0.1	66.71 ± 0.8	(Dy, Tb) Fe_2_	1490	
	37.29 ± 0.4	55.31 ± 0.8	7.40 ± 1.0	hcp (Dy, Tb)		
4#-Dy_15_Tb_15_Fe_70_	11.45 ± 0.6	11.81 ± 0.5	76.74 ± 0.6	(Dy, Tb) Fe_3_	1494, 1501	L → (Dy, Tb) Fe_3_ L → (Dy, Tb) Fe_2_ + (Dy, Tb) Fe_3_
	14.57 ± 0.1	17.95 ± 0.1	67.48 ± 0.1	(Dy, Tb) Fe_2_		
5#-Dy_13.5_Tb_13.5_Fe_73_	10.34 ± 0.3	12.91 ± 0.2	76.75 ± 0.1	(Dy, Tb) Fe_3_	1475, 1505	
	13.07 ± 0.3	17.41 ± 0.3	69.52 ± 0.6	(Dy, Tb) Fe_2_		
6#-Dy_12_Tb_12_Fe_76_	3.96 ± 0.3	6.37 ± 0.5	89.67 ± 0.4	(Dy, Tb)_2_ Fe_17_	1482, 1495, 1555	L → (Dy, Tb)_6_ Fe_23_ L→ (Dy, Tb) Fe_3_ + (Dy, Tb)_6_ Fe_23_
	9.05 ± 0.1	11.73 ± 0.3	79.22 ± 0.3	(Dy, Tb)_6_ Fe_23_		
	11.01 ± 0.2	12.95 ± 0.3	76.04 ± 0.4	(Dy, Tb) Fe_3_		
7#-Dy_10.5_Tb_10.5_Fe_79_	3.54 ± 0.1	6.16 ± 0.5	90.30 ± 0.4	(Dy, Tb)_2_ Fe_17_	1497, 1562	
	9.14 ± 0.1	11.48 ± 0.2	79.38 ± 0.2	(Dy, Tb)_6_ Fe_23_		
	10.10 ± 0.2	13.33 ± 0.2	76.57 ± 0.4	(Dy, Tb) Fe_3_		
8#-Dy_9_Tb_9_Fe_82_	3.54 ± 0.1	5.18 ± 0.3	91.28 ± 0.4	(Dy, Tb)_2_ Fe_17_	1523, 1560	L → (Dy, Tb)_2_ Fe_17_ L→ (Dy, Tb)_2_ Fe_17_ + (Dy, Tb)_6_ Fe_23_
	10.21 ± 0.1	11.87 ± 0.4	77.92 ± 0.5	(Dy, Tb)_6_ Fe_23_		

**Table 2 materials-16-04697-t002:** Thermodynamic parameters of the Dy-Tb-Fe system.

Phase	Thermodynamic Parameters	Reference
Liquid	LFe,DyLiquid=−32,524.791+10.7T0	[24]
	LFe,DyLiquid=14,150.219−1.17T1	[24]
	LFe,DyLiquid=422.595−8.2T2	[24]
	LFe,TbLiquid=−20,115.512−3.2T0	[24]
	LFe,TbLiquid=−3332.5193−11.1T1	[24]
	LFe,TbLiquid=660.3−8.1T2	[24]
	LDy,Tb,FeLiquid=60000	This work
bcc (α-Fe, δ-Fe, β-Dy, β-Tb)	LFe,Dybcc=38,0000	[24]
	LFe,Tbbcc=58,0000	[24]
fcc (γ-Fe)	LFe,Dyfcc=38,0000	[24]
	GDyfcc0=GDyhcp0+5000	[24]
	LFe,Tbfcc=75,0000	[24]
	GTbfcc0=GTbhcp0+5000	[24]
hcp (α-Dy, α-Tb)	LFe,Dyhcp0=100,000	[24]
	LFe,Tbhcp0=150,000	[24]
Fe_17_RE_2_ (Fe_17_Dy_2_, Fe_17_Tb_2_)	GmFe17Dy2=−12,882+3.477T+0.1053G0Dyhcp+0.8947G0Febcc	[24]
	TcFe17Dy2=371,β0Fe17Dy2=0.161	[24]
	GmFe17Tb2=−10,343+1.668T+0.1053G0Tbhcp+0.8947G0Febcc	[24]
	TcFe17Tb2=408,β0Fe17Tb2=0.167	[24]
	LDy,Tb:FeFe17RE20=−2000	This work
Fe_23_RE_6_ (Fe_23_Dy_6_, Fe_23_Tb_6_)	GmFe23Dy6=−15,700+3.909T+0.2069G0Dyhcp+0.7931G0Febcc	[24]
	TcFe23Dy6=534,β0Fe23Dy6=0.1	[24]
	GmFe23Tb6=−12,501+0.355T+0.2069G0Tbhcp+0.7931G0Febcc	[24]
	TcFe23Tb6=574,β0Fe23Tb6=0.01	[24]
	LDy,Tb:FeFe23RE60=−2000	This work
Fe_3_RE (Fe_3_Dy, Fe_3_Tb)	GmFe3Dy=−15,506+3.101T+0.25G0Dyhcp+0.75G0Febcc	[24]
	TcFe3Dy=606,β0Fe3Dy=0.493	[24]
	GmFe3Tb=−12,008−0.277T+0.25G0Tbhcp+0.75G0Febcc	[24]
	TcFe3Tb=652,β0Fe3Tb=0.426	[24]
	LDy,Tb:FeFe3RE0=−3000	This work
Fe_2_RE (Fe_2_Dy, Fe_2_Tb)	GmFe2Dy=−17,860−5.419T−0.00672T2+3R105+Tln1−exp−210T	[24]
	TcFe2Dy=635,β0Fe2Dy=0.89	[24]
	GmFe2Tb=−14,540−4.332T−0.0074T2+3R1852+Tln1−exp−185T	[24]
	TcFe2Tb=704,β0Fe2Tb=0.762	[24]
	LDy,Tb:FeFe2RE0=−4000	This work

## Data Availability

All the data that support the findings of this study are included within the article.

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
