# Peer review of "Solidification Behavior of Dy-Tb-Fe Alloys through Experimental Study and Thermodynamic Calculation"

_materials, 2023, doi:10.3390/ma16134697_

Round 1
Reviewer 1 Report
The article is devoted to the study of the microstructure of solidification and phase transitions of Dy-Tb-Fe alloy samples using various analysis methods. In general, this direction is quite interesting, since the results obtained allow not only to solve a number of technological issues related to the production of ternary systems, but also to gain fundamental knowledge in this area of research. The presented results are in good agreement with each other and the literature data, and the article itself is written in a fairly understandable and high-quality presentation of the results. This work corresponds to the subject of the declared journal and can be accepted for publication after the authors answer a number of questions related to their interpretation of the data obtained.
1. The introduction is too short and uninformative in the presented form, the authors use too many references to the same facts (up to 7–8 references per sentence), in this regard, it is proposed to expand the introduction and provide more analysis of currently known similar alloys and their prospects.
2. The results of energy dispersive analysis should be given with lower accuracy, as well as measurement error, since with small changes, the results of the error may reflect the absence of these changes.
3. The authors should explain about the data presented in Figures 2-6. The diffraction pattern was obtained from a massive sample or from a specific location containing an inclusion. It should also be noted that the presented diffraction pattern contains reflections with a strongly distorted shape. Therefore, the authors should describe these effects in more detail.
4. Authors should give the weighted contributions of each observed phase. As well as their structural parameters and the dynamics of their change.
5. The technical remarks include the poor quality of the presented figures of the phase diagrams, they should be significantly improved, since it is impossible to evaluate the observed effects in the presented form.
Author Response
Response to the reviewers
The authors thank the reviewers for the helpful comments and suggestions. In the following, these comments and questions are addressed in detail.
Reviewer #1:
The article is devoted to the study of the microstructure of solidification and phase transitions of Dy-Tb-Fe alloy samples using various analysis methods. In general, this direction is quite interesting, since the results obtained allow not only to solve a number of technological issues related to the production of ternary systems, but also to gain fundamental knowledge in this area of research. The presented results are in good agreement with each other and the literature data, and the article itself is written in a fairly understandable and high-quality presentation of the results. This work corresponds to the subject of the declared journal and can be accepted for publication after the authors answer a number of questions related to their interpretation of the data obtained.
Q1. The introduction is too short and uninformative in the presented form, the authors use too many references to the same facts (up to 7-8 references per sentence), in this regard, it is proposed to expand the introduction and provide more analysis of currently known similar alloys and their prospects.
Answer: Thanks for the comments. The related expressions were revised carefully and were marked by highlight color in the text. Please check them in the revised manuscript.
Q2. The results of energy dispersive analysis should be given with lower accuracy, as well as measurement error, since with small changes, the results of the error may reflect the absence of these changes.
Answer: Thanks for the comments. During the EDS measurements, the phase compositions of each phase in alloy samples were tested four times at different field. Now the standard deviations of these data were given in Table 1, We revised the related expressions in the text. Please check them in the revised manuscript.
Q3. The authors should explain about the data presented in Figures 2-6. The diffraction pattern was obtained from a massive sample or from a specific location containing an inclusion. It should also be noted that the presented diffraction pattern contains reflections with a strongly distorted shape. Therefore, the authors should describe these effects in more detail.
Answer: Thanks for the comments. The diffraction patterns as shown in Figures 2-6 were obtained from metal powders, which were made from as-cast alloy samples. The background of the diffraction patterns (seen in Fig. 2(b)) is too high, and the spectral peaks in some diffraction patterns (seen in Fig. 6(b)) are not smooth, and, which could result from the stress and/or preferential orientation produced through the preparation of metal powders. The preparation of as-cast Dy-Tb-Fe powder alloys is not easy due to the oxidation of rare earth metals. In addition, the crystallization of as-cast alloys is not perfect, leading to their poor diffraction patterns. We revised the related expressions in the text. Please check them in the revised manuscript.
Q4. Authors should give the weighted contributions of each observed phase. As well as their structural parameters and the dynamics of their change.
Answer: Thanks for the comments. The refinements of the XRD patterns of the as-cast Dy-Tb-Fe alloy samples were not performed in this work because the quality of XRD patterns of the as-cast Dy-Tb-Fe alloy samples is not high due to the poor crystallization of as-cast alloys. The phase structure parameters of the formed phases in the alloys samples were not determined. Furthermore, the volume fraction of each phase in each alloy sample was not analyzed through the XRD patterns. Actually, the volume fraction of each phase in each alloy sample was also analyzed by the SEM images using the picture software. These results were not given in the text due to the large errors of the images analysis.
Q5. The technical remarks include the poor quality of the presented figures of the phase diagrams, they should be significantly improved, since it is impossible to evaluate the observed effects in the presented form.
Answer: Thanks for the comments. The quality of all the figures in the manuscript is improved, and we believe, they are enough clear now in the text. Please check them in the revised manuscript.
Reviewer 2 Report
The manuscript is a good contribution to the field of Dy-Tb-Fe alloys with thermodynamic information and phase characterization with X-ray diffraction and differential thermal analysis.
Some minor considerations:
The experimental procedure does not provide details about the casting system and its cooling conditions that generate the samples. This information is extremely important to make the work reproducible and also to confront the Scheil-Gulliver application hypothesis. The authors should discuss why the Scheil-Gulliver model can be applied to the solidification conditions imposed by the casting device.
Plase clarify whether the samples used for the XRD analyses are in powder or bulk form. If they are in bulk, discuss whether there is texture interference with the results.
State the criteria used to choose the nine sample compositions shown in Table 1.
Please include in the reference the related work and its relevant information:
"Microstructural studies of ternary rare earth-iron alloys", Journal of the Less Common Metals, Volume 126, December 1986, Pages 107-112. https://doi.org/10.1016/0022-5088(86)90255-9
Author Response
Response to the reviewers
The authors thank the reviewers for the helpful comments and suggestions. In the following, these comments and questions are addressed in detail.
Reviewer #2:
The manuscript is a good contribution to the field of Dy-Tb-Fe alloys with thermodynamic information and phase characterization with X-ray diffraction and differential thermal analysis. Some minor considerations:
Q1. The experimental procedure does not provide details about the casting system and its cooling conditions that generate the samples. This information is extremely important to make the work reproducible and also to confront the Scheil-Gulliver application hypothesis. The authors should discuss why the Scheil-Gulliver model can be applied to the solidification conditions imposed by the casting device.
Answer: Thanks for the comments. The cooling conditions of as-cast alloy samples were shown in the text. In our experiments, the alloy samples were cooled in a copper crucible filled with cooling water. The cooling rate is very fast, and thus the solidification of the as-cast alloy samples is non-equilibrium process. The Scheil-Gulliver model is to simulate the non-equilibrium solidification. During the non-equilibrium solidification simulated by the Scheil-Gulliver model, it is assumed that the solute diffuses uniformly and rapidly in the liquid phase, but the diffusion cannot occur in the solid phase. In this work, the Scheil-Gulliver model is used to simulate the non-equilibrium solidification process of the as-cast alloy samples. The related expressions were revised carefully in the text. Please check them in the revised manuscript.
Q2. Plase clarify whether the samples used for the XRD analyses are in powder or bulk form. If they are in bulk, discuss whether there is texture interference with the results.
Answer: Thanks for the comments. The XRD measurements were performed using powder alloy samples in this work. The background of the diffraction patterns (seen in Fig. 2(b)) is too high, and the spectral peaks in some diffraction patterns (seen in Fig. 6(b)) are not smooth, and, which could result from the stress and/or preferential orientation produced through the preparation of metal powders. The preparation of as-cast Dy-Tb-Fe powder alloys is not easy due to the oxidation of rare earth metals. In addition, the crystallization of as-cast alloys is not perfect, leading to their poor diffraction patterns. The related expressions in the ‘Experimental procedure’ section were revised in the text. Please check them in the revised manuscript.
Q3. State the criteria used to choose the nine sample compositions shown in Table 1.
Answer: Thanks for the comments. In order to examine the primary phase formed in the Dy-Tb-Fe ternary system, A series of alloy samples along with the section of xDy : xDy=1 :1 were designed and parapared in our experiments.
Q4. Please include in the reference the related work and its relevant information: "Microstructural studies of ternary rare earth-iron alloys", Journal of the Less Common Metals, Volume 126, December 1986, Pages 107-112. https://doi.org/10.1016/0022-5088(86)90255-9
Answer: Thanks for the comments. This reference was cited in the text.
Round 2
Reviewer 1 Report
The authors answered all the questions, the article can be accepted for publication.